# Evaluation of the Effects of Different Phosphorus Sources on *Microcystis aeruginosa* Growth and Microcystin Production via Transcriptomic Surveys

Zihao Li, Lili An, Feng Yan, Wendi Shen, Wenjun Du and Ruihua Dai *

Department of Environmental Science and Engineering, Fudan University, Shanghai 200433, China; 20210740070@fudan.edu.cn (Z.L.); 22210740001@m.fudan.edu.cn (L.A.); 21210740032@m.fudan.edu.cn (F.Y.); 20210740027@fudan.edu.cn (W.S.); 21210740044@m.fudan.edu.cn (W.D.)
* Correspondence: rhdai@fudan.edu.cn; Tel.: +86-21-5566-4354

**Abstract:** *Microcystis aeruginosa* (*M. aeruginosa*) is a dominant species among frequent cyanobacterial blooms and is well known for its toxin-producing ability. Phosphorus (P) is a typical growth-limiting element for *M. aeruginosa*. Although alterations in physiological reactions have been documented in response to various P sources, the underlying molecular processes and transcriptional patterns remain poorly understood. This study evaluated the physiological and molecular responses of *M. aeruginosa* to different P sources. The growth of *M. aeruginosa* was promoted by both dissolved inorganic phosphorus (DIP) and dissolved organic phosphorus (DOP) at a concentration of 0.4 mg/L with an initial cell density of $1.50 (\pm 0.05) \times 10^6$ cells/mL. The cell density reached $1.38 (\pm 0.05) \times 10^7$ cells/mL in the DIP group on day 14, a value which was higher than that in the DOP group. Most photosynthesis genes had higher levels of upregulated expression in the DIP group. For instance, gene *psbA* was upregulated by 0.45 $Log_2$Fold Change ($Log_2$FC). In the DOP group, it is interesting that the Pi ($PO_4$-P) concentration increased to 0.09 mg/L on day 14. Meanwhile, the expression of the gene encoding alkaline phosphatase-like protein was significantly upregulated, with a value of 1.56 $Log_2$FC, and the alkaline phosphatase concentration increased in the DOP group. The intracellular microcystin (IMC) concentration decreased with time in both groups. However, the concentration of extracellular microcystins (EMCs) increased with incubation time in both groups. Phosphorus participates in the regulation of microcystin synthesis, mainly by regulating ATP synthesis. Based on the physiological and molecular investigations in this study, the results provide crucial insights into the physiological adaptations and the role of P in modulating harmful algal bloom formation, microcystin synthesis, and potential molecular responses of *M. aeruginosa*.

**Keywords:** *Microcystis aeruginosa*; dissolved inorganic phosphorus; dissolved organic phosphorus; transcriptomic; microcystins

## 1. Introduction

Eutrophication is induced by nitrogen and phosphorus loadings in aquatic systems. High levels of eutrophication in aquatic systems are typically associated with harmful algal blooms (HABs). HABs are rapid growths of cyanobacteria that can cause remarkable damage to aquatic ecosystems [1,2]. Accordingly, their proliferation has become a global environmental issue and has gained considerable attention [3]. *Microcystis aeruginosa* (*M. aeruginosa*), one of the most widespread bloom-forming cyanobacteria, mainly develops in organic-rich freshwater bodies such as lakes and ponds. Microcystins (MCs), which are cyclic heptapeptide compounds generated by *M. aeruginosa*, are extremely hepatotoxic [4,5]. Controlling HABs and toxin generation have been priorities of research globally, with critical implications for human health.

Prior studies on the effects of nitrogen on *M. aeruginosa* have been relatively comprehensive and have covered various forms of nitrogen, nitrogen starvation, and nitrogen

sources at different concentrations [6,7]. Additionally, the effects of these on an organism's physiological state and transcriptome level have been deemed relatively sufficient [7,8]. Moreover, a previous study reported that *M. aeruginosa* exhibited the highest growth rate and maximal optical density when cultured in an N:P ratio of 0.1 [9], indicating that the growth of *M. aeruginosa* may be stimulated by an excessive amount of nitrogen and a consistent level of phosphorus in the culture medium. Phosphorus (P) is an essential nutrient for the development and metabolism of cyanobacteria as it is required for the formation of ribosomes, DNA, and cell membranes [10,11]. Furthermore, the availability of phosphorus has an important impact on cyanobacterial growth and metabolism and is regarded as a limiting factor for the occurrence of cyanobacterial blooms [12,13]. To date, most investigations, ranging from field to gene expression studies, have focused on the role of phosphorus in promoting cyanobacterial blooms and toxicity. Accordingly, physiological changes and MC production based on molecular processes through transcriptomic analysis in response to different phosphorus sources have not been well studied.

Among all the phosphorus compounds in natural waters, dissolved inorganic phosphorus (DIP) and dissolved organic phosphorus (DOP) are the dominant forms [14–16]. Zhang et al. discovered that the growth of *M. aeruginosa* was enhanced by elevated concentrations of DIP, but inhibited at high concentrations (0.6 and 1.0 mg P/L) of DOP after P starvation. The high concentration of DIP was found to promote the production of MCs in *M. aeruginosa*, while the high concentration of DOP induced the release of intracellular MCs without affecting MC production [17]. DIP was inferred to be a more bioavailable phosphorus source because *M. aeruginosa* growth was promoted more significantly by DIP than by DOP. Cyanobacteria always bloom during the summer with a large uptake of phosphorus, thereby leading to DIP depletion in freshwater [18–20]. The DIP content in surface water is extremely low, causing widespread phosphorus stress or even starvation [10,20]. To overcome P stress or starvation during this period, *M. aeruginosa* may shift to assimilate DOP to survive and remain the dominant species in natural waters. However, these conditions may trigger physiological variations and affect toxin production mechanisms. Studies regarding the cellular responses in *M. aeruginosa* under the condition that DOP serves as the sole P source may help evaluate toxin production and toxin persistence in cyanobacterial blooms. Moreover, further assessments can provide basic knowledge that will improve the phosphorus control of cyanobacteria in natural waters.

Numerous studies have determined the effects of various phosphorus sources on cyanobacteria. Li et al. evaluated the effects of three different phosphorus compounds, sodium-β-glycerophosphate (β-gly), adenosine triphosphate (ATP), and orthophosphate ($K_2HPO_4$), on *M. aeruginosa* [21]. The outcomes of their study demonstrated that *M. aeruginosa* can utilize both DIP and DOP. However, the intracellular responses of *M. aeruginosa* to different forms of P remain unknown. Fluorescence quantitative PCR was performed to identify those transcripts of the target genes related to P transportation (*sphX*, *pstS,* and *phoX*) and MC synthesis (*mcyA* and *mcyD*) and in which the *pstS* gene was found to be expressed with the highest abundance in the lowest concentration (0.02 mg/L) in the DOP group [17]. The effects of DIP and DOP on phosphorus uptake and toxin synthesis by *M. aeruginosa* were examined at the molecular level. However, understanding the expressions of several of these genes is insufficient because cyanobacterial metabolism is regulated by multiple genes. To fully comprehend how *M. aeruginosa* responds to DIP and DOP, the genes that are differentially expressed must be identified. Transcriptomics enables the investigation of gene transcription and gene expression at the RNA level. The response of cyanobacteria to different phosphorus sources can be more accurately assessed if genetic changes are combined with physiological changes. To develop innovative strategies to control cyanobacterial blooms, the growth, P metabolism, and toxicity of *M. aeruginosa* under DIP and DOP must be understood.

This study investigated the response of *M. aeruginosa* to different phosphorus sources through transcriptome analysis. Incubations were performed to evaluate the development of *M. aeruginosa* on DIP and DOP substrates. Furthermore, the changes in P concentrations

were monitored, and the variations in P utilization were compared. The transcriptome data were compared with the physiological evaluation data of cell density, chlorophyll-a (chl-a), photosynthesis, MC production, and alkaline phosphatase (AP) concentration. This study aimed to (1) determine the effects of two different phosphorus sources on the growth of *M. aeruginosa*, (2) reveal the molecular mechanism of P utilization, MC production, and AP activity in *M. aeruginosa*, and (3) improve our understanding of the relationship between MC production and P. This study elucidates the physiological responses of *M. aeruginosa* to different phosphorus sources at the molecular level.

## 2. Materials and Methods

### 2.1. Microcystis aeruginosa and Cultures

*M. aeruginosa* (FACHB-905), isolated from cyanobacterial blooms in Dianchi Lake in China, was obtained from the Institute of Freshwater Algae Culture Collection at the Institute of Hydrobiology, Chinese Academy of Sciences (Wuhan, China). *M. aeruginosa* was cultured and grown in BG11 medium [22]. All solutions were prepared with deionized water. *M. aeruginosa* was cultivated in a climatic chamber at 25 °C with a light intensity of 27 μmol m$^{-2}$ s$^{-1}$ on a 12:12 light/dark cycle (BIC-300, Boxun, Shanghai, China). All flasks were gently shaken by hand three times daily to avoid cell aggregation or settlement [7].

### 2.2. Experimental Design

*M. aeruginosa* cells in the exponential growth phase were collected via centrifugation ($3800\times g$ for 10 min), washed three times with sterilized deionized water, and inoculated in the sterilized phosphorus-free BG-11 medium for 7 days to exhaust intracellular phosphate. Subsequently, the P-free medium was divided into two different groups: (1) one that used dipotassium phosphate medium ($K_2HPO_4$) as DIP and (2) one that used disodium beta-glycerophosphate pentahydrate medium (β-gly, containing C-O-P bonds) as DOP, with both containing 0.4 mg/L P, the maximum total phosphorus concentration required for Class V water according to China's surface water environmental quality standards [23]. The control group was *M. aeruginosa* medium on Day 0. The initial cell biomass was adjusted to 1.5 ($\pm$0.05) $\times 10^6$ cells/mL, and *M. aeruginosa* was inoculated in 1 L Erlenmeyer flasks. All experiments were done in triplicates.

The experimental period lasted for 14 days. The cell density, chl-a concentration photosynthetic activity, and concentration of total phosphorus (TP) and $PO_4$-P (Pi) were measured every day at the same time until day 8 and then measured every other day, whereas MCs and AP were analyzed every two days.

### 2.3. Physiological Measurements

2.3.1. Measurement of Cell Density and Growth Rate

A UV-vis spectrophotometer (L6S, Leng-Guang, Shanghai, China) was used to measure the optical density at 680 nm ($OD_{680}$) [24], which was found to be positively associated with cell numbers. The specific growth rate (μ) was calculated as per Equation (1):

$$\mu = (\ln N_2 - \ln N_1)/(t_2 - t_1) \tag{1}$$

where $N_1$ and $N_2$ are the cell densities at times $t_1$ and $t_2$, respectively.

2.3.2. Extraction and Analysis of Chlorophyll a

10 mL cell culture was collected and centrifuged at $3600\times g$ for 10 min, and the supernatant was discarded. The cells were rinsed with 5 mL of 90% acetone and left overnight at 4 °C in darkness. Then, the extracts were filtered with a 0.45 μm filter membrane, and absorbances at 630, 645, 663, and 750 nm were measured using a UV–vis

spectrophotometer (L6S, Leng-Guang, Shanghai, China) to calculate the content of chl-a according to Equations (2) and (3), with a 90% acetone solution used as the blank [25].

$$\text{chl-a (mg/L)} = [11.64 \times (OD_{663} - OD_{750}) - 2.16 \times (OD_{645} - OD_{750}) + 0.10 \times (OD_{630} - OD_{750})] \cdot V_1 V^{-1} \delta^{-1} \tag{2}$$

$$\text{chl-a (pg/cell)} = (2) \times 10^9 / \text{cell density} \tag{3}$$

Here, $V_1$ is the constant volume of the extracted cultures (mL), V denotes the sample volume, and $\delta$ is the optical path of the cuvette.

### 2.3.3. Analysis of Photosynthetic Activity

To evaluate photosynthetic activity, the Fv/Fm ratio was calculated. A 4-mL portion of the cell culture was dark-incubated for 15 min. The Fluorometer (AquaPen AP 100, Aqualabo, Brno, Czech Republic) was used to monitor the cells' photosynthetic activity by measuring the variable-to-maximum chlorophyll fluorescence ratio (Fv/Fm ratio).

### 2.3.4. Measurement of Total Phosphorus and $PO_4$-P

Subsamples of the DIP and DOP were filtered with a 0.45 μm filter. According to the instructions for the use of the Hach total phosphorus reagent, 5 mL of supernatants were added in the reagent tube, then potassium persulfate was added and digested at 150 °C for 30 min. After the mix was cooled to room temperature, 2 mL of NaOH and a packet of phosver3 phosphate reagent were added to the tube. The sample was used to determine the levels of TP by using the ammonium molybdate spectrophotometric method, and the undigested supernatants were used to determine the levels of Pi by using the Mo-Sb Anti-Spectrophotometer in the culture [26]. TP and Pi concentration were measured with an ultraviolet-visible spectrophotometer (HACH, DR6000).

### 2.3.5. Extraction and Measurement of Microcystins

The samples were centrifugated at $3800 \times g$ for 10 min. After centrifugation, the supernatant was collected to assess extracellular microcystin (EMC) levels, and the cyanobacterial cells were utilized in order to identify the intracellular microcystins (IMCs). The cell pellet was frozen at $-20$ °C for 24 h [7]. The IMCs were extracted by using the freeze-thaw method, wherein the cyanobacterial cells were freeze-thawed three times [17]. MC concentrations were evaluated using an enzyme-linked immunosorbent assay (ELISA) kit (Product No.062964, Shanghai MLBIO Biotechnology Co., Ltd., Shanghai, China) and detected through Microplate Reader–Spectra Max M2 (Molecular Devices, 3860 N First Street, San Jose, CA, Canada). The required operating methods were followed exactly as instructed by the kit [27].

### 2.3.6. Extraction and Analysis of Alkaline Phosphatase

The supernatant was collected to determine AP concentration via an enzyme-linked immunosorbent assay (ELISA) kit (Product No. 062906, Shanghai MLBIO Biotechnology Co., Ltd., Shanghai, China) and also detected through Microplate Reader–Spectra Max M2 (Molecular Devices, 3860 N First Street San Jose, CA, Canada). The absorbance was measured at a 450-nm wavelength, and the content of AP in the sample was calculated by using a standard curve.

### 2.4. RNA Extraction and Sequencing

For transcriptomic analysis, the samples were collected in 30 mL centrifuge tubes and rapidly harvested via centrifugation at $2500 \times g$ for 10 min at 4 °C. The samples were immediately frozen in liquid N after harvesting and stored at $-80$ °C until sequencing. All sample procedures were conducted within 30 min. To minimize temporal variations in gene expression and cell physiology, cultures were collected at the same time every day for all groups. All groups were analyzed in triplicate [28]. Total RNA was extracted

using TRIzol reagent (Invitrogen Life Technologies, Carlsbad, CA, USA). The mRNA was isolated using a Ribo-Zero rRNA Removal Kit (Illumina, San Diego, CA, USA) according to the manufacturer's instructions. A NanoDrop spectrophotometer (NC2000, Thermo Scientific, Waltham, MA, USA) and a Bioanalyzer 2100 system (Agilent, Santa Clara, CA, USA) were used to assess RNA quality and integrity. The abundances of the transcripts were measured in terms of RPKM (Reads Per Kilobase per Million reads) for the analysis of the transcriptome data. DESeq (version 1.18.0) tools were used to perform a differential expression analysis for comparisons between samples with three biological replicates. The metabolic pathways were mapped using the Kyoto Encyclopedia of Genes and Genomes (KEGG) database. The Microcystis DEGs were mapped to the NCBI (National Center for Biotechnology Information) genome database for KEGG analysis.

## 3. Results and Discussion

### 3.1. Growth and Photosynthetic Activity of Microcystis aeruginosa

3.1.1. Physiological Changes in Growth and Photosynthesis

Cell density and chl-a concentration act as indicators of growth, while the Fv/Fm ratio is used to evaluate the photosynthetic activity of *M. aeruginosa*. As shown in Figure 1A,B, *M. aeruginosa* growth was promoted under both phosphorus conditions, but the cell density was higher in the DIP group than in the DOP group. In the DIP group, the specific growth rate remained high in the first 10 days and then steadily decreased over the next several days. As shown in Figure 1B, the specific growth rate in the DIP group was higher than that in the DOP group after day 4, consistent with the higher cell density in the DIP group. The maximum cell density reached 1.38 ($\pm$0.05) $\times 10^7$ cells/mL by the end of the experiment. In the DOP group, *M. aeruginosa* maintained a stable specific growth rate, and the cell density on the last day was 0.74-fold that in the DIP group. This result is consistent with that reported by Yuan et al. [29], who showed that *M. aeruginosa* grows better in the presence of $K_2HPO_4$ as the phosphorus source than in the presence of β-gly and ATP. However, a previous study found that the maximal cell density of *M. aeruginosa* in DOP groups was only 15% of that in the DIP groups under an initial concentration of 0.6 mg P/L [17]. This apparent difference may be attributed to the different initial phosphorus concentration settings and experimental periods. DIP uptake stimulates the growth and cell division of *M. aeruginosa*, leading to the formation of large blooms. In contrast, DOP uptake stimulates the production of extracellular enzymes, which can cleave DOP into inorganic forms. This can increase the availability of DIP and promote the further growth of *M. aeruginosa* [30].

Chl-a concentrations are a key marker of photosynthetic activity in *M. aeruginosa* [6]. As shown in Figure 1C, the chl-a content increased significantly from day 0 to day 10 and then showed a slight increase in both groups, peaking at 0.62 pg/cell on day 10 in the DIP group and 0.57 pg/cell on day 10 in the DOP group. The change in chl-a content displayed a similar trend to that of cell density, aligning with previous findings [7]. These results indicate that both DIP and DOP can stimulate the photosynthetic activity of *M. aeruginosa*. It is worth noting that the increase in chl-a content was slightly higher in the DIP group compared with that in the DOP group; this may be attributed to the higher bioavailability of DIP.

Fv/Fm values are positively correlated with photosystem (PS) II performance, which acts as a photosynthetic efficiency parameter [31]. As illustrated in Figure 1D, compared to the initial Fv/Fm value (0.12), the maximum values corresponded to 0.40 (DIP) and 0.36 (DOP), showing a significant increase in photosynthetic efficiency in both groups. Previous studies reported that the Fv/Fm values in DIP ($K_2HPO_4$) groups were higher than those in DOP (β-gly) groups under the condition of 0.2 mg P/L [17], in alignment with our findings, indicating higher cell density in DIP groups. The Fv/Fm values of both groups increased and showed gradual declines until the end of cultivation, thereby aiding cyanobacterial growth and proliferation.

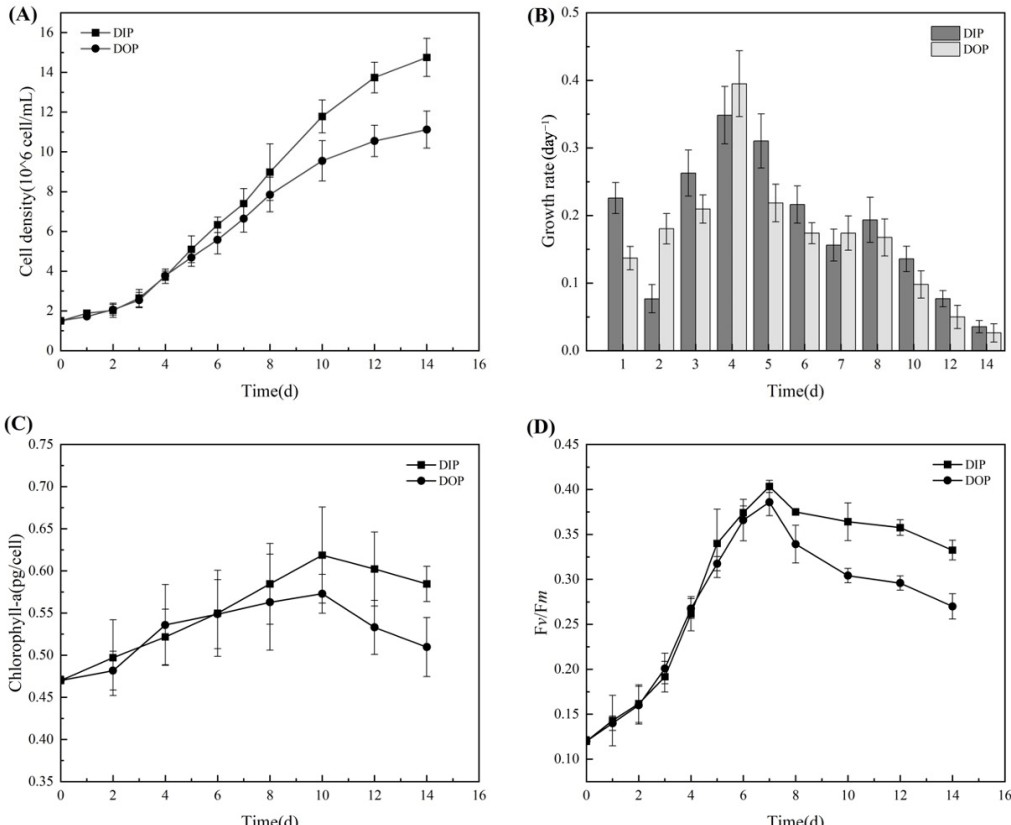

**Figure 1.** Changes in cell density (**A**), specific growth rate (**B**), and chlorophyll-a (**C**) and Fv/Fm (**D**) of the *M. aeruginosa* with time under DIP and DOP conditions. The error bars represent the standard errors.

Our results showed that the growth changes in cell density displayed similar patterns in both DIP and DOP groups, and that the growth rate first increased and then decreased. Besides, when DOP was used as the sole P source, the chl-a contents and the Fv/Fm ratio were marginally lower than they were in the DIP group, indicating that DOP was bioavailable to *M. aeruginosa* and could help *M. aeruginosa* alleviate DIP depletion.

3.1.2. Changes in Photosynthetic Genes' Response to Different Phosphorus Sources

The transcriptional abundance of most genes involved in photosynthesis was variably expressed after the addition of different phosphorus sources. According to the literature, both PS I and PS II take part in the conversion of light energy into the energy of chemical bonds [32]. As shown in Figure 2A, compared to what was observed in the control, most PS I genes in the DIP group showed significant changes, and the expression of *psaC* was markedly elevated by 1.39 $\text{Log}_2\text{Fold Change}$ ($\text{Log}_2\text{FC}$). PS I gene expression in the DOP group exhibited a similar pattern, with an elevated expression range of 0.13 to 0.38 $\text{Log}_2\text{FC}$. Most of the detected genes involved in PS II, such as *psbC*—which encodes the PS II chlorophyll-binding protein CP43—and *psbU*—which encodes the PS II 12 kDa extrinsic protein [33]—had upregulated transcript abundance in the DIP group. However, other genes, such as *psbY*, *psbZ*, *psb27*, and *psb28*, exhibited a reduction in relative abundance. For example, *psb28*, one of the assembly factors of PS II [34], was downregulated by 0.19 $\text{Log}_2\text{FC}$ on day 5. Moreover, approximately half of the PS II-related genes were upregulated in the DOP group: a number fewer than that of the PS II-related genes in the DIP group. This result was consistent with the value of Fv/Fm in the DOP group (Figure 1D), which was lower than that in the DIP group.

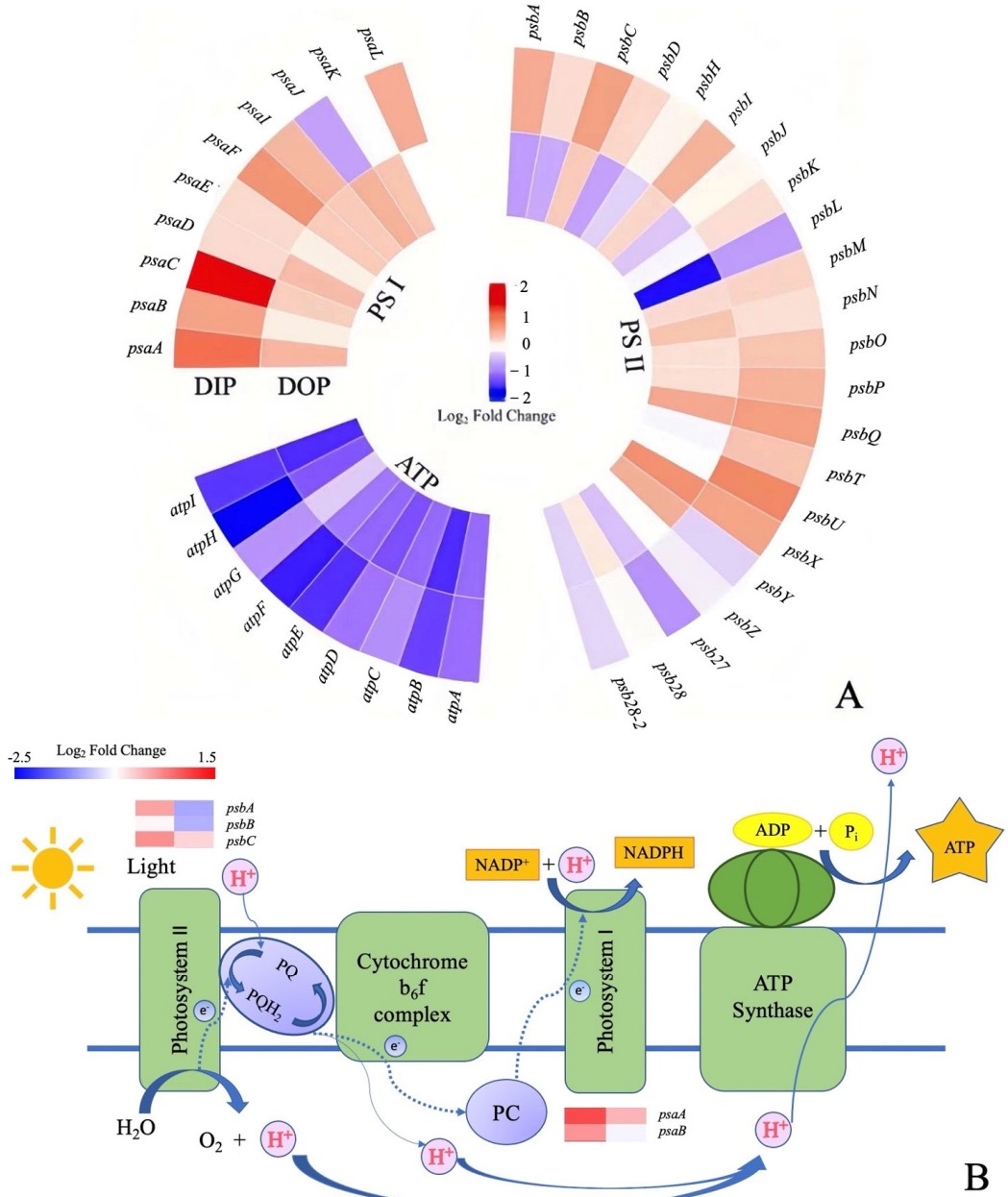

**Figure 2.** Circle heat map of gene expression of genes involved in photosynthesis (**A**) and photosynthesis on the thylakoid membrane of *M. aeruginosa* with key genes involved in PS I and PS II (**B**). PQ: plastoquinone; PQH$_2$: plastid hydroquinone; PC: plastocyanin. A reduction in transcript abundance corresponds to a blue hue, whereas an increase in transcript abundance corresponds to a red color. White indicates that there is no difference between the experimental and control conditions.

All genes encoding ATP synthase subunits showed reductions in expression, as shown in Figure 2A. In the DIP group, the expressions of *atpA* (encoding ATP synthase F1 and alpha subunit), *atpB* (encoding ATP synthase F1 and beta subunit), and *atpI* (encoding ATP synthase I family protein) were downregulated by 1.17, 1.49, and 1.56 Log$_2$FC, respectively. Similarly, in the DOP group, the transcript abundances of these three genes (*atpA*, *atpB*, and *atpI*) were downregulated by 1.18, 1.64, and 1.63 Log$_2$FC, respectively. These results indicate that the energies of photosynthesis in both DIP and DOP group were lower than that in the control. This might be due to the phosphorus supply in the DIP and DOP groups being severely depleted, resulting in decreased ATPase activity in both groups. The

cyanobacteria in the DIP and DOP groups had both decreased their energy consumption by reducing their photosynthesis activity.

Cyanobacteria are the principal producers in freshwater environments and are considered ideal model organisms for photosynthesis studies owing to their high photosynthetic efficiency and rapid development rate. The photosynthetic apparatus of cyanobacteria mainly consists of PS II, cytochrome $b_6f$, PS I, phycobilisomes, and ATP synthase [35], as shown in Figure 2B. In PS II, the gene *psbA*, which encodes the photosynthetic reaction center protein [36], was upregulated by 0.45 $Log_2FC$ in the DIP group. However, this gene was downregulated by 0.85 $Log_2FC$ in the DOP group, as shown in Figure 2A. A previous report revealed that the dinoflagellate Peridinium bipes, a typical component of the phytoplankton group, can form blooms in freshwater ecosystems, and that the expression of *psbA* in the DOP (β-gly, 52 μmol/L) groups was downregulated compared to that in the DIP ($K_2HPO_4$, 52 μmol/L) groups, ultimately aligning with the findings of our investigation [37]. In addition, the expressions of *psbC*, which encodes light-harvesting proteins in PS II [38], were similar between the DIP and DOP groups. Accordingly, rather than being connected with the activity of the light-harvesting protein complex, the PS II photoreactive central protein of cyanobacteria was more active in the DIP group, and may have resulted in greater cell development in the DIP group.

As shown in Figure 2B, PS I is the second PS involved in the photo-cooperative photoreaction of cyanobacteria. A full membrane protein complex called PS I uses light energy to produce the high-energy carriers ATP and NADPH [39]. The large subunits PsaA and PsaB, which are the core subunits in PS I, are located in the center of the PS I monomer, and encode the PS I P700 chlorophyll-a apoproteins A1 and A2, respectively. Moreover, most chlorophyll and carotenoids in the antenna system are found in subunits PsaA and PsaB [33]. Therefore, the photosynthetic activity of cyanobacteria was markedly influenced by the expression of *psaA* and *psaB*. In this experiment, both the DIP and DOP groups displayed considerably high expression of *psaA*, which was upregulated by 0.88 $Log_2FC$ in the DIP group and 0.36 $Log_2FC$ in the DOP group. Although both genes were upregulated, the fold change in the DIP group was greater than that in the DOP group. In addition, the upregulated expression of *psaB* in the DIP group was markedly higher than that in the DOP group. Overall, the cells in the DIP group grew substantially more quickly than those in the DOP group owing to variations in the expression of genes involved in photosynthesis.

Our results indicated that both phosphorus sources promoted the upregulated expression of photosynthetic genes in *M. aeruginosa*. Additionally, we found that the gene encoding the expression of the PS II photoreactive central protein was significantly differentially expressed, compared to other genes, between the DIP and the DOP groups, leading to the concentration of chl-a in the DIP group being higher than that in the DOP group.

*3.2. Variations in Related Genes' Responses to Dissolved Inorganic Phosphorus and Dissolved Organic Phosphorus*

3.2.1. Utilization of Different Phosphorus Sources and Alkaline Phosphatase Activity

The assimilation of P by *M. aeruginosa* was indicated by the concentrations of TP and Pi in the culture medium. The initial TP concentration in both groups was 0.4 mg/L. As shown in Figure 3A, after six days of inoculation, the concentration declined to 0.08 mg/L and fluctuated slightly on subsequent days in the DIP group. In contrast to the rapid reduction in TP concentrations in the DIP group, TP concentrations declined slowly and subsequently remained stable over the study period when β-gly was employed as a P source. The decreasing TP concentrations in both DIP and DOP groups indicate that DIP and DOP can be utilized by *M. aeruginosa* in natural water to facilitate its growth [29]. Li et al. reported that after P starvation, when *M. aeruginosa* utilized DIP ($K_2HPO_4$) and other DOPs (β-gly, ATP), TP in the DIP group decreased sharply during the first 3 days of incubation and remained low regardless of the initial phosphorus concentration (0.1, 1.0, 2.0, or 4.0 mg/L) [21]. The results demonstrate that DIP was more rapidly assimilated than DOP and was used directly by *M. aeruginosa*.

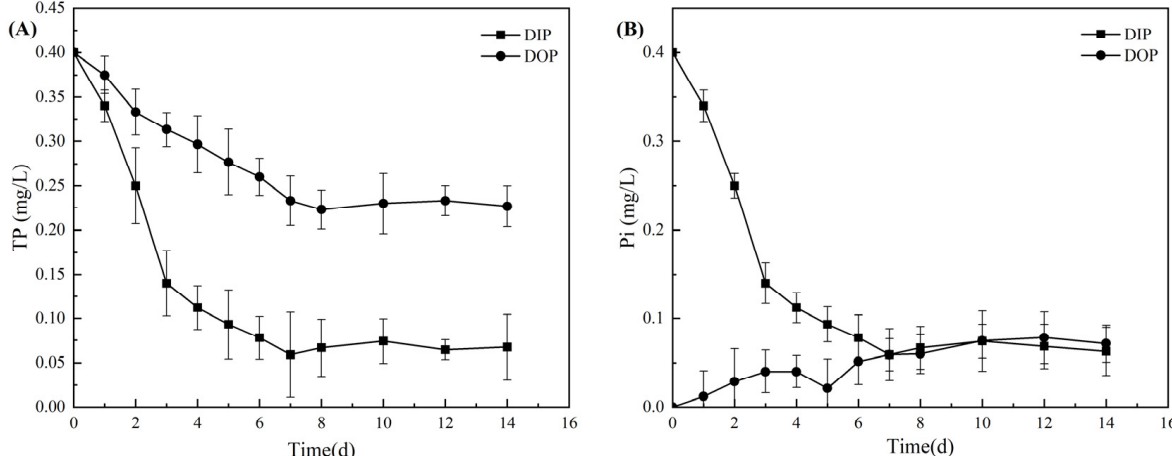

**Figure 3.** Changes in phosphorus concentration under different phosphorus source media with time. (**A**) TP concentrations of both groups; (**B**) Pi concentrations of both groups.

Figure 3B shows the variations in Pi concentrations in both groups. Interestingly, a minor quantity of Pi was detected in the DOP group. The concentration of Pi grew steadily, reaching 0.09 mg/L at the end of cultivation. This result agreed with that of a previous study [17], which found that Pi concentrations increased to 0.03 and 0.11 mg P/L at 0.2 and 0.6 mg P/L in the DOP (β-gly) groups. The Pi concentration of the DOP (β-gly) groups was reported to peak at 9.217 mg/L on day 8 with *Nostoc* sp., in comparison to the concentration of 2 mg/L on the first day [40]. A possible reason is that AP, a critical enzyme of *M. aeruginosa*, converts complex organophosphates to phosphate; this may explain the observed increase in Pi concentration in DOP group over time [41].

Figure 4 shows the variations in AP concentrations. AP is crucial in order for cyanobacteria to utilize phosphate esters of all organic phosphorus sources, such as phosphate monoesters and phosphate diesters [42]. *M. aeruginosa* grown in the absence of P has been found to have slower growth rates and higher AP concentration [43]. As shown in Figure 4, AP was detected in the supernatant of the cyanobacterial culture after phosphorus deprivation for one week (0.8 ng/mL, day 0). When *M. aeruginosa* was inoculated in the DIP medium, the AP levels declined quickly to a low level on day 2 and remained stable for the following days at $0.15 \pm 0.05$ ng/mL. In contrast, the AP concentration increased at the beginning of cultivation in the DOP group. On day 6, the maximal concentration of AP in the β-gly groups reached 1.25 ng/mL, a value which was higher than that of the initial AP. This result suggests that *M. aeruginosa* may have a higher capacity to hydrolyze DOP compounds into Pi under conditions of DIP limitation. Therefore, DOP may represent an important phosphorus source for cyanobacterial growth in aquatic ecosystems, especially under DIP limitation.

The results demonstrated that DIP was a preferred P source for *M. aeruginosa* because the P was depleted faster in the DIP group. Moreover, we discovered that the AP concentration in the DIP group was much lower than that in the DOP group. The AP concentration increased slightly in the DOP group, possibly because *M. aeruginosa* required more AP, which might hydrolyze extracellular DOP and convert it into Pi for absorption and utilization by *M. aeruginosa*.

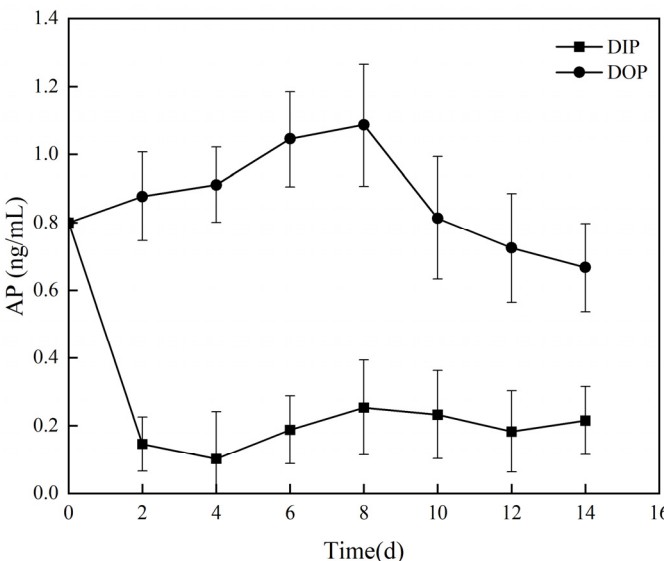

**Figure 4.** Variations of Alkaline phosphatase concentration.

### 3.2.2. Variations in Related Genes Encoding Phosphorus Metabolism under Different Phosphorus Sources

Cyanobacteria thrive in a permanently changing aquatic environment, undergoing seasonal challenges and possibly overcoming inorganic phosphate deficiency. These conditions trigger physiological variations, through methods such as boosting the hydrolysis of organophosphorus by adjusting intracellular storage or using exogenous dissolved organophosphorus to activate multiple phosphorus usage pathways. AP is an enzyme that is essential for phosphorus metabolism because it releases phosphate from complex organophosphate compounds [44]. Figure 5A shows a heatmap of the genes related to P sensing, transportation, and AP. An unsymbolized gene that regulates an AP-like protein was discovered in this study, with its expression abundance exceeding 1.56 $Log_2FC$ in the DOP group. This result may indicate that this gene regulates the synthesis of substances involved in the hydrolysis of *β-gly* in order to be assimilated by *M. aeruginosa*. When DOP accounts for a significant proportion of the total dissolved P, the bioavailability of DOP to *M. aeruginosa* may serve as a potential reason that leads *M. aeruginosa* to become a dominant species [21,30].

Pst is an ATP-dependent ABC transporter-type system that includes phosphate-binding and transmembrane protein subunits [45]. As shown in Figure 5A, gene *pstS* is involved in phosphate ABC transporter substrate-binding protein, the expression of which was upregulated by 0.55 $Log_2FC$ in the DIP group. However, *pstS* expression was downregulated by 0.08 $Log_2FC$ in the DOP group. The transcript abundance of *pstS* has also been found to be lower in the *β-gly* groups than in the DIP groups in *Raphidiopsis raciborskii* [46]. *PstA* and *PstC* are transmembrane subunits that form a channel in the inner membrane. Both genes had similar expression levels in the DIP and DOP groups, indicating that the expression of these genes was not significantly affected by different phosphorus sources. Furthermore, the expression of *pstB*, which is involved in the phosphate transport system ATP-binding protein, was upregulated by 0.27 and 0.05 $Log_2FC$ in the DIP and DOP groups. In this study, the expression of related genes was generally upregulated when the Pst system responded to different phosphorus sources. The expression difference between *pstS* and *pstB* was more significant, indicating that phosphate-binding protein activity (*PstS* and *PstB*) was more affected than transmembrane protein activity (*PstA* and *PstC*) in the DIP group.

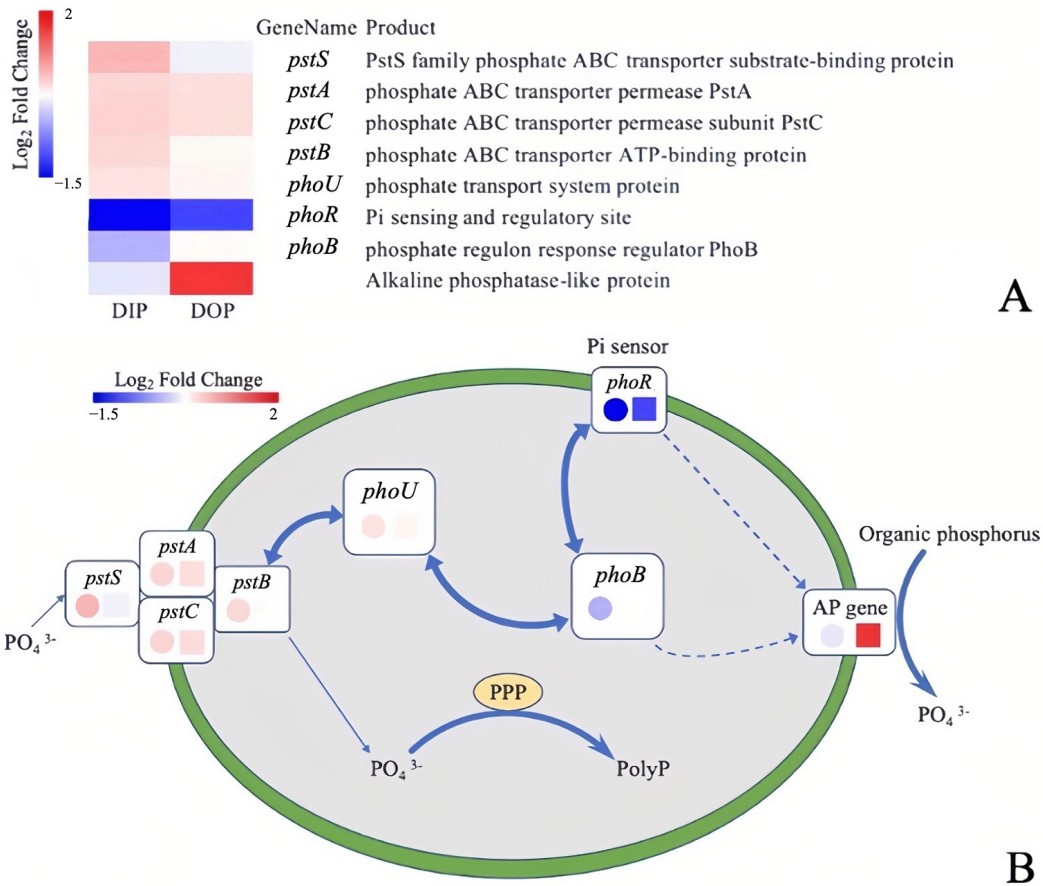

**Figure 5.** The heatmap of genes involved in P transportation and AP activity (**A**) and the schematic diagram of phosphorus transportation under DIP and DOP and metabolic pathways potentially involved in the utilization of different P sources (**B**). DIP and DOP are represented by circles and squares, respectively. Reductions in transcript abundance correspond to blue hues, whereas increases in transcript abundance correspond to red colors. White indicates that there is no difference between the experimental and control conditions. PPP: polyphosphatase.

An overview of the phosphate transport mechanism is presented in Figure 5B, where four proteins (encoded by *PstS*, *PstC*, *PstA*, and *PstB*) constitute an ABC transporter in the Pst system required for high-affinity phosphate capture in the cytoplasm [47]. *phoU*, *phoR*, and *phoB* were found to play key roles in phosphate sensing and regulation, as shown in Figure 5B. PhoU, a regulatory protein that interacts with PstB and PhoB, regulates Pi import via the Pst system. The expression of *phoU* was upregulated by 0.19 and 0.07 $Log_2FC$ in the DIP and DOP groups, respectively. Previous studies revealed that the PhoU protein can sense a conformational change in the Pst system when Pi is imported [48], indicating that both phosphorus sources can be absorbed by cyanobacteria as Pi. PhoR is the Pi-sensing site, and its expression was downregulated by 1.50 and 1.09 $Log_2FC$ in the DIP and DOP groups, respectively. Moreover, the expression of the phosphate regulon response regulator *phoB* was downregulated by 0.43 $Log_2FC$ in the DIP group. The PhoB/R two-component system was induced by Pi starvation [49]. Therefore, the expression of these two genes in the DIP group was lower than that in the control group and appeared to be reasonable. However, the expression of *phoB* was upregulated by 0.04 $Log_2FC$ in the DOP group, and this might be due to the regulatory effect of *phoU*. In addition to coordinating Pi input, *phoU* may affect the expression of *phoB* [50], thereby affecting the expressions of part of the AP genes in PhoB regulators. The AP genes receive information on Pi stress from the Pi sensor PhoR or PhoB regulators and then hydrolyze the organophosphorus to Pi. Therefore, genes related to AP were obviously upregulated in the DOP group.

The corresponding hydrolysis and utilization pathways in cyanobacteria are shown in Figure 5B. In the presence of adequate phosphorus, cyanobacteria absorb extra phosphorus and store it in their cells. According to previous research on the cyanobacterial proteome and transcriptome, cyanobacterial cells store phosphorus as polyphosphate in phosphorus-rich environments [51]. Phosphorus is stored as polyphosphate by polyphosphatases (PPP), allowing cells to simultaneously employ phosphorus from polyphosphate and convert ATP to ADP.

### 3.3. Effects of Different Phosphorus Sources on Microcystins Production

The variations in IMC contents under different phosphorus sources are shown in Figure 6A. The concentrations of IMCs followed a similar trend in both the DIP and DOP groups, initially increasing and then decreasing. The greatest level of IMC was achieved on the second day, with 51.1 and 53.8 µg/L in the DIP and DOP groups, respectively. The initial increase in IMC content could be attributed to the stimulation of microcystin production by the availability of phosphorus. After day 8, IMC content in the two groups declined and remained stable. By day 10, the IMC content decreased to 23.9 µg/L in the DIP group. This result is consistent with that of Zhang et al. [17], who demonstrated that IMC concentrations in the DIP and DOP groups significantly decreased over time (on day 18 compared to day 8).

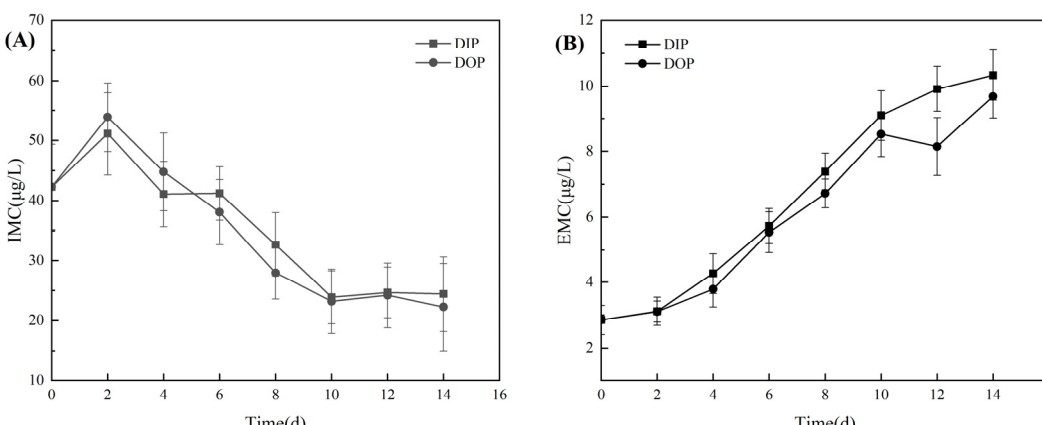

**Figure 6.** Variations in MCs under a different form of P groups. (**A**) IMC (µg/L) and (**B**) EMC (µg/L). The error bars represent the standard errors.

As shown in Figure 6B, the EMC content displayed an upward trend. The EMC contents of the DIP and DOP groups increased until they reached their peak values on day 14. The peak concentrations of EMC in the DIP and DOP groups were 10.34 and 9.68 µg/L, respectively. These values indicate 5.59- and 3.40-fold higher concentrations than the initial concentrations. Considering Figure 1A, it is clear that the changes in EMC concentration aligned with the cyanobacterial biomass variations, which increased with time in both groups. This conclusion aligns with that of earlier studies, which showed that EMC levels were compatible with the growth rates of cyanobacterial cells [17]. In addition, the lysis and the death of cyanobacterial cells cause the release of IMCs into the surrounding environment, which increases EMCs [52].

### 3.4. Relation between Microcystins Biosynthesis and Phosphorus

Previous research linked the synthesis of MCs to nitrogen in the environment [53] because nitrogen is a component of MCs. Although phosphorus is not a component of MCs, most studies have discovered that phosphorus considerably impacts *M. aeruginosa* proliferation and MC production [54,55].

Phosphorus is essential for trapping, storing, transferring, and distributing energy via high-energy phosphate bonds [56]. Phosphorus potentially participates in metabolic

processes to control MC production, as shown in Figure 7. DOP must first be hydrolyzed by AP secreted by *M. aeruginosa*, and then Pi can be taken up into the cell through the phosphate transport system [57]. DIP, on the other hand, can be taken up directly by *M. aeruginosa* through membrane-bound phosphate transport proteins [58]. Pi was then assimilated to support *M. aeruginosa* metabolism. *M. aeruginosa* has a sophisticated regulatory system for the assimilation of phosphorus. Thylakoid membranes can mediate fully functional photosynthetic and respiratory electron transport within the photosynthetic lamellae of cyanobacteria [59]. Phosphorus is absorbed to create ATP, which provides energy for MC synthesis and hence regulates enzyme performance and the MC production pathway. Additionally, polyphosphate (polyP), which serves as a nutrient store for cyanobacterial cells, may be formed from ATP by polyphosphate synthase [60]. In summary, phosphorus as an energy source plays a major role in MC synthesis.

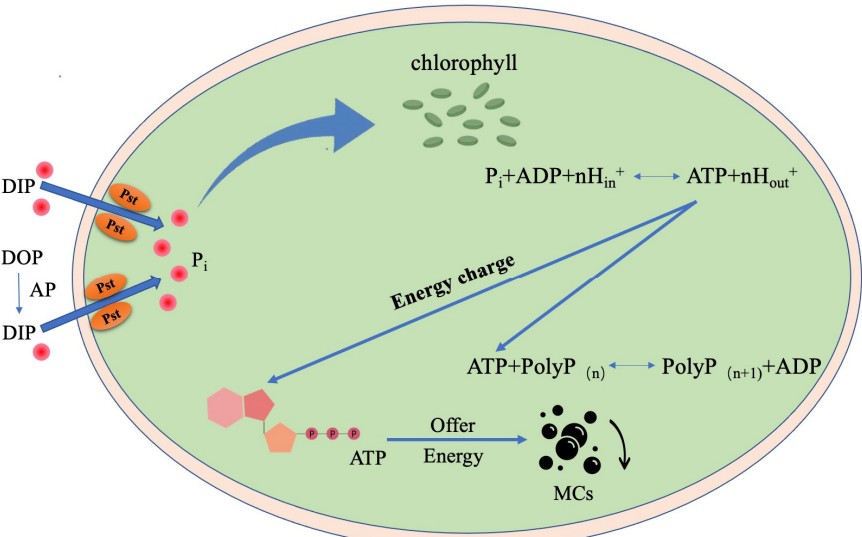

**Figure 7.** Phosphorus potentially participates in metabolic processes that control the production of MCs.

In this study, most key genes encoding F-type ATP synthase were significantly down-regulated in the DIP and DOP groups (Figure 2A), possibly decreasing ATP levels and influencing intracellular energy metabolism and hence affecting MC synthesis. These results align with the observation that IMC concentrations decreased with incubation time. Overall, these findings imply that phosphorus may influence the metabolism of *M. aeruginosa* and regulate MC production by affecting adenylate energy charge. Further, the MC contents and energy charge had a direct relationship, and MC synthesis was determined by energy charge. Our results, at the molecular level, align with those of the previous study [61]. Phosphorus is a crucial component in the production of ATP, which is required in order to provide energy for the synthesis of MCs. Compared to the effects of different phosphorus sources on genes involved in MC synthesis, this might be the primary mechanism by which phosphorus governs the production of MCs [17].

Improving our understanding of MCs' gene expression and regulation will help us identify their potential cellular functions. Enzymes encoded by the *mcy* gene cluster play a crucial role in the biosynthesis of MCs. The MCs' synthase gene cluster comprises 10 genes and is transcribed as two polycistronic transcripts (*mcyABC* and *mcyDEFGHIJ*) [62]. The relative expressions of the detected genes involved in MC production (*mcyABCDEGJ*) are shown in Figure 8. In this study, *mcyD* expression was downregulated by 0.12 $Log_2FC$ in the DIP group and 0.20 $Log_2FC$ in the DOP group. *mcyD* is reported to be essential for MC synthesis because it is involved in the creation of the amino acid Adda (2S,3S,8S,9S)-3-amino-9-methoxy-2,6,8-trimethyl-10-phenyldeca-4,6-dienoicacid, which is one of the key components of MCs [63]. Based on our findings, the downregulated expression of *mcyD* may be the primary cause of the final decline in IMC contents in cyanobacterial cells,

supporting the result of Anam et al. [64], who discovered that *mcyD* gene expression was consistent with MC contents. Increased transcription levels of the microcystin-related gene *mcyA* in *M. aeruginosa* could lead to an elevated production of MCs [65]. The expression of *mcyA* was upregulated by 0.19 Log$_2$FC in the DIP group, consistent with the findings of a previous study [66]. Meanwhile, the *mcyA* expression was downregulated by 0.29 Log$_2$FC, possibly having led to lower IMC contents in the DOP group compared to in the DIP group from day 6. Furthermore, *mcyJ* is known to be a tailoring gene, putatively involved in O-methylation [62], and its expression was slightly upregulated by 0.19 and 0.09 Log$_2$FC in the DIP and DOP groups, respectively, indicating that the synthesis activity of MCs was higher in the DIP group. The differential expression of *mcyA* and *mcyJ* may have resulted in higher concentrations of both EMC and IMC in the DIP group. Additionally, in both DIP and DOP groups, no appreciable differences were found in the expressions of some genes involved in peptide synthesis, such as *mcyBCEG*. Although DIP is considered a more bioavailable P source, DOP often accounts for most of the ambient P pool in aquatic ecosystems. Further, an outbreak of cyanobacterial blooms of *M. aeruginosa* is more likely to occur and be sustained in lakes and reservoirs with DOP being predominant [67]. In the future, managing eutrophication and lowering MC production from cyanobacterial blooms will require consideration of DOP in aquatic environments.

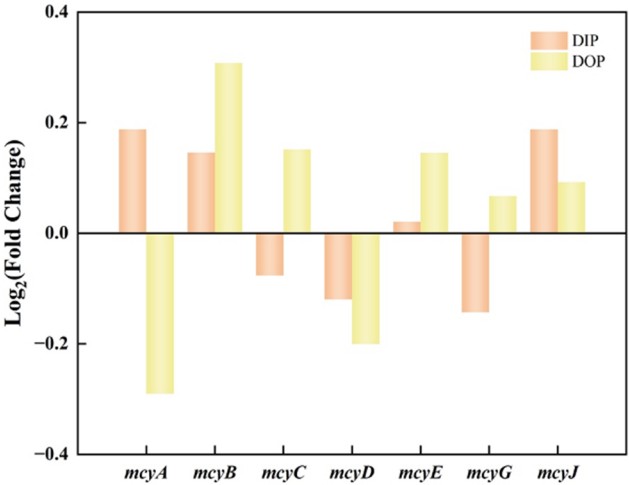

**Figure 8.** Variations of the *mcy* gene expressions in the clusters of the DIP and DOP groups.

## 4. Conclusions

DIP and DOP both promoted the growth of *M. aeruginosa*, and the values of the chl-a content and Fv/Fm in the DIP group were higher than those in the DOP group, indicating that DIP has a more evident influence on the proliferation of cyanobacteria. The transcriptomic results revealed that the variations in most PS I and PS II genes were slightly different. The PS II central protein PsbA was more frequently active in the DIP group; the difference in *PsbA* expression might affect photosynthesis, which would, in turn, affect cell density. Notably, the AP synthesis-related genes were significantly more abundant in the DOP group than that in the DIP group, indicating that *M. aeruginosa* could absorb phosphorus by hydrolyzing extracellular DOP by AP at the molecular level. IMC concentrations first increased and then decreased with time. The expressions of most *mcy* genes showed a little difference between the DOP and DIP group. The expression of *mcyD*, the key gene encoding MC synthesis in the *mcy* cluster, was downregulated in both DIP and DOP groups. This indicates that P controls ATP production, which indirectly controls MC synthesis. These results emphasize the complexities of cyanobacterial cell physiology and related gene expression in the presence of different P sources and provide vital insights into the role of P in controlling HAB generation and toxin production.

**Supplementary Materials:** The following supporting information can be downloaded at: https://www.mdpi.com/article/10.3390/w15101938/s1, Table S1: NPvsKD5.DESeq; Table S2: NPvsBGPD5.DESeq.

**Author Contributions:** Z.L.: investigation, data curation, formal analysis, visualization, writing—original draft, writing—review & editing. L.A.: investigation, formal analysis. F.Y.: investigation, data curation. W.S.: investigation, validation. W.D.: investigation. R.D.: funding acquisition, writing—review & editing, conceptualization. All authors have read and agreed to the published version of the manuscript.

**Funding:** This research was supported by the National Natural Science Foundation of China (No. 51678159).

**Data Availability Statement:** In supplementary materials.

**Acknowledgments:** The author is extremely appreciative of the teacher's advice and the assistance of the research group's pupils.

**Conflicts of Interest:** The authors declare no conflict of interest.

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
