# Peer review of "Evaluation of the Effects of Different Phosphorus Sources on Microcystis aeruginosa Growth and Microcystin Production via Transcriptomic Surveys"

_water, doi:10.3390/w15101938_

Round 1
Reviewer 1 Report
Phosphorus is a typical growth-limiting element for M. aeruginosa, the molecular responses mechanisms of M. aeruginosa to different type of phosphorus. In this manuscript, the growth and transcriptomic analysis of algal cell to dissolved inorganic phosphorus (DIP) and dissolved organic phosphorus were screened. This work is interesting and valuable data was provided. In my opinion, it worthy to be published after minor revision.
1. Data is enough, but the analysis is weak and should be strenghtnened.
2. Effect of DOP and DIP on growth are significant different and determined by many different integrated metabolic pathways, please illustrate the mechanism deeply.
3. Role of DOP and DIP in cellular metabolism is very different and affects many aspects of the cell besides alkaline phosphatase, microcystin synthesis and algal growth. furthermore analysis should be conducted to illustrate the effects.
4. If you seperate the discussion from the results, it would be better to do deeply discussion.
Some sentences need to be polished.
Reviewer 2 Report
Report on the manuscript entitled (
Evaluation of the effects of different phosphorus sources on Microcystis aeruginosa growth and microcystin production via transcriptomic surveys) submitted to the Water is quite popular research and has been dealt by many authors previously. However, there is always a scope to review and crystallize new information for the overall benefaction of the scientific society working in this direction. Therefore, this manuscript requires additional details and essential corrections, and then it will be re-submitted to Water.
1- The topic is timely and worth revision. However, the paper lacks a coherent structure and in its present form is merely a compilation of information without liason and critical analyses.
2- Half of the abstract is an introductory paragraph and the authors should include more quantitative results and the structure of the abstract should be considered
3- The authors measured the Measurement of cell density as optical density spectrophotometrically at 680nm. Most of the references for measuring the optical density of the cyanobacteria were done over 780 nm.
4- For the first time appearing the abbreviations should be mentioned in the full name and afterwards it can be abbreviated. Also, avoid abbreviations in the titles and subtitles.
5- The quality of the figures is very poor and should be improved.
6- The Relation between MC biosynthesis and P and genes regulation are good results, however, the authors did not discuss this relationship in the discussion
7- The references should be updated at least the last 3 yeras
8- Overall the article is a good piece of work and can be helpful for readers in the field of phycology and health.
Minor improvement required
Reviewer 3 Report
This is a review of a manuscript entitled ‘Evaluation of the effects of different phosphorus sources on Microcystis aeruginosa growth and microcystin production via transcriptomic surveys’ presented for publication in Water (MDPI). The manuscript describes the impact of phosphorus source on basic physiology, microcystin production and transcriptome of Microcystis aeruginosa.
I think that the manuscript must be improved before publication.
Below are the points that I find most important to be considered by the Authors before reconsideration of the manuscript.
1. Expand introduction. Elaborate on phosphorus impact on M. aeruginosa growth, nitrogen-to-phosphorus ratio, published gene expression studies in different phosphorus conditions, proteomics data. You have the first transcriptomic data on DIP/ DOP source in M. aeruginosa samples and you have to present them in the light of the previous findings, however limited they might be.
2. Line 153: what kind of digestion? With what enzyme? What conditions?
3. You do not actually do experiments in triplicates to assure accuracy, but to be able to do some statistics, to estimate standard deviation etc. In my opinion it is enough to write ‘experiments were done in triplicates’ without further explanations.
4. Lines 274-275: DIP and DOP are misplaced in the text.
5. Figure 2: cytochrome b6f complex is usually denominated with lower case letters.
6. You compare your photosystems data with published plants data. Cyanobacteria and plants have similar, but not the same subunit compositions of both PSI and PSII. There are structural data available for cyanobacterial PSs so these should be used. The result of this is, for example, that you write about PSI being present in non-stacked regions of thylakoids (line 240), while in most cyanobacteria thylakoids do not form stacks at all.
7. Fv/Fm is a normalized measure of quantum efficiency of PSII. It is not dependent on the amount of PSII centres in the sample, only on their ability to utilize light to drive photosynthesis. Based on this it is not correct to draw relations between the degree of PSII genes expression upregulation and Fv/Fm values. Fv/Fm values can be used to assess general fitness of present PSII centres and entire photosynthetic electron transport chain, but not assess the number of PSII centres in the system.
8. Lines 366-368: the claim here is a bit strong for the scarcity of data from other species. Please elaborate more (if you have stronger data), or weaken the claim.
9. Lines 370-371: this needs clarification. As I understand you are trying to say that the PstS is a phosphate ABC transporter substrate-binding protein. It is not evident from the phrase written as it is now.
10. You describe transcriptomic data. This means that the information you have is on the transcript level only. However, in the text you often use protein abbreviations (i.e. expression of PstS) or terms such as ‘expression of protein’. Both are not specific enough to be sure what kind of information you actually have. You should be more careful in such claims and make sure to stick to transcripts, and gene names throughout the manuscript text.
11. This might be a personal preference but I would not call cyanobacteria algae, because they are prokaryotic and algae are generally considered eukaryotic organisms. I guess ‘microalgae’ is correct and simply ‘cyanobacteria’ is the best. Using the word ‘algae’ allows you to compare M. aeruginosa to completely unrelated eukaryotic organisms.
12. Line 434: you claim to have higher EMC concentrations in DIP group but it is not supported by the data in Fig. 6B. Error bars clearly show that there are no statistical differences. You might have a trend but not a clear increase in one compared to the other.
13. Fig. 7: as I understand microcystin is only produced by cyanobacteria. Cyanobacteria are prokaryotic, so they do not have any organelles surrounded by membranes, hence no chloroplasts. There clearly is a chloroplast in figure 7, although described as thylakoid in figure caption.
14. Several times in the manuscript you refer to a control group (e.g. lines: 251, 255, 397). What was the control group? It is not described in the methods chapter. What kind of phosphorus source did the control group have?
15. The manuscript title suggest that the reader will be presented with transcriptomic data. But in the manuscript only information on limited genes is given. It would be preferable if the Authors presented at least a list of all identified transcripts as a supplementary table.
The quality of English language is mediocre. There are no spelling mistakes, but in some cases the choice of the words is odd. There is also one sentence that is hard to understand, that I addressed in the Comments and Suggestions for Authors. I advise to use a professional language editor.
Round 2
Reviewer 2 Report
I have gone through the MS and I found that the authors considered all comments that I have raised in my corrections
Author Response
Thank you for providing helpful comments to improve our work. We greatly appreciate your efforts in reviewing the manuscript.
Reviewer 3 Report
Revised manuscript entitled 'Evaluation of the effects of different phosphorus sources on Microcystis aeruginosa growth and microcystin production via transcriptomic surveys' was presented for consideration.
Authors very kindly responded to my questions. Manuscript is presented in improved form. Apart from few things pointed out in Language section of the review, I only see one substantial error in the text:
-Lines 271-273 'PS I contains specialized reaction center and antenna proteins that convert light energy into chemical energy, while PS II contains a complex of proteins and cofactors that split water and generate oxygen' This is not true. Both PSI and PSII take part in photosynthetic electron transport chain - in other words: in conversion of light energy into energy of chemical bonds. Additionally, there is an oxygen evolving complex associated with PSII, that provides electrons for photosynthetic electron transport chain.
After this and language errors are corrected I think the manuscript will be fit for publication in Water.
English language is of good quality. The only things I have to point out are:
- Line 311 'were' is used instead of 'was'.
- Word 'algae' was replaced by 'cyanobacteria' in the whole text without taking care of whether the hrase need 'cyanobacteria' or 'cyanobacterial'. This must be corrected.
- Line 412 'bioavailability of M. aeruginosa to DOP' - shouldn't it be the other way around? bioavailability of DOP to M. aeruginosa?
- Unify the way you write 'upregulation'. There are both 'upregulation' and 'up-regulation' in the text.
-Lines 536-537 'A crucial step of MCs biosynthesis is involved by the enzymes encoded by the mcy gene cluster' - this sentence must be corrected.
-Bacterial gene names comprise of three lower case letters followed by a capital letter. Authors should stick to this general rule.
